# The enterococcal cytolysin synthetase has an unanticipated lipid kinase fold

Shi-Hui Dong[1†], Weixin Tang[2†], Tiit Lukk[3], Yi Yu[1], Satish K Nair[1,4]*, Wilfred A van der Donk[1,2]*

[1]Department of Biochemistry, University of Illinois at Urbana-Champaign, Urbana, United States; [2]Roger Adams Laboratory, Department of Chemistry, Howard Hughes Medical Institute, University of Illinois at Urbana-Champaign, Urbana, United States; [3]Cornell High Energy Synchrotron Source, Ithaca, United States; [4]Center for Biophysics and Computational Biology, University of Illinois at Urbana-Champaign, Urbana, United States

**Abstract** The enterococcal cytolysin is a virulence factor consisting of two post-translationally modified peptides that synergistically kill human immune cells. Both peptides are made by CylM, a member of the LanM lanthipeptide synthetases. CylM catalyzes seven dehydrations of Ser and Thr residues and three cyclization reactions during the biosynthesis of the cytolysin large subunit. We present here the 2.2 Å resolution structure of CylM, the first structural information on a LanM. Unexpectedly, the structure reveals that the dehydratase domain of CylM resembles the catalytic core of eukaryotic lipid kinases, despite the absence of clear sequence homology. The kinase and phosphate elimination active sites that affect net dehydration are immediately adjacent to each other. Characterization of mutants provided insights into the mechanism of the dehydration process. The structure is also of interest because of the interactions of human homologs of lanthipeptide cyclases with kinases such as mammalian target of rapamycin.

*For correspondence: s-nair@life.uiuc.edu (SKN); vddonk@illinois.edu (WAD)

†These authors contributed equally to this work

## Introduction

Cytolysin is produced by many clinical isolates of *Enterococcus faecalis* and consists of two post-translationally modified peptides termed cytolysin L and S (*Figure 1A*) (*Gilmore et al., 1994*; *Cox et al., 2005*). These peptides have lytic activity against various types of eukaryotic cells including immune cells (*Cox et al., 2005*; *Bierbaum and Sahl, 2009*). The production of cytolysin enhances virulence in infection models of *E. faecalis*, and epidemiological data support an association with acute patient mortality (*Ike and Clewell, 1984*; *Huycke et al., 1991*; *Chow et al., 1993*; *Van Tyne et al., 2013*). Cytolysin is a member of the lanthipeptides, a family of polycyclic peptides that are made in a two-step process involving dehydration of Ser and Thr residues to dehydroamino acids and subsequent addition of thiols of Cys residues to the dehydroamino acids (*Figure 1B*) (*Knerr and van der Donk, 2012*). This process, catalyzed by the enzyme CylM for cytolysin, generates the characteristic thioether crosslinks called lanthionine (Lan) and methyllanthionine (MeLan) (*Figure 1*). An N-terminal leader peptide in the substrates is important for substrate binding by lanthipeptide biosynthetic enzymes (*Oman and van der Donk, 2010*), but the post-translational modifications take place in the C-terminal core peptides.

To date, four distinct routes to lanthipeptides have been discovered, illustrating that the cyclic thioether motif is a privileged structural scaffold that has been independently accessed multiple times during evolution (*Zhang et al., 2012*). The thioether bridges introduce conformational constraints that facilitate target binding and reduce proteolytic susceptibility. The four routes differ primarily in the mechanism of dehydration. For the class I, III, and IV lanthipeptides, the mechanism of dehydration

**eLife digest** *Enterococcus faecalis* is a bacterium that is usually found living harmlessly in the gut of humans and other mammals. However, over the past few decades hospitals have noted an increase in the number of hospital-acquired infections caused by antibiotic-resistant strains of *E. faecalis*.

Many of the *E. faecalis* strains that cause illness and death do so by producing a toxin called cytolysin, which can destroy a range of cells, including the immune cells that normally eradicate bacterial infections. Inside the bacteria, an enzyme called cytolysin synthetase—also known as CylM—catalyzes the reactions that make the cytolysin toxin from precursor molecules.

Enzymes are primarily made up of proteins. Both the sequence of the amino acids in the protein chains and the shapes and structures that these chains fold into affect how the enzyme works. CylM is made up of two parts, or 'domains'. One of these, known as the dehydration domain, removes water molecules from some of the precursor amino acid chains that are used to build cytolysin. This dehydration reaction forms the first stage of cytolysin production. How CylM catalyzes this reaction was not known, because CylM does not have a similar amino acid sequence to any other enzymes and no information about its structure was available.

Now, Dong, Tang et al. have resolved the structure of the *E. faecalis* CylM enzyme using a technique called x-ray crystallography. Unexpectedly, this revealed that the dehydration domain of the enzyme has a similar structure—despite having a completely different amino acid sequence—to enzymes that are found in eukaryotic organisms (i.e., organisms with cells that contain a nucleus). These enzymes are called lipid kinases, and help to add phosphate groups to other molecules.

Additional structural and biochemical analyses enabled Dong, Tang et al. to investigate how CylM catalyzes the dehydration reaction in more detail. Given its central role in toxin production, an increased understanding of how CylM makes cytolysin could eventually help to develop new treatments for the conditions caused by *E. faecalis* infections.

has been illuminated by crystallographic characterization of the dehydratases or close sequence homologs (*Li et al., 2006, 2007*; *Goto et al., 2010*; *Ortega et al., 2015*), but the mechanism of dehydration for class II lanthipeptide synthetases (LanMs) that include CylM has remained enigmatic. These enzymes show no clear sequence homology with non-lanthipeptide proteins and despite two decades of investigation, structural information on class II lanthipeptide synthetases has been unavailable. Such information would be valuable for obtaining inhibitors of cytolysin biosynthesis that could be therapeutically valuable. In addition, LanM lanthipeptide synthetases are involved in the biosynthesis of several lanthipeptides and their derivatives that are under clinical evaluation such as actagardine and duramycin (*Grasemann et al., 2007*; *Steiner et al., 2008*; *Jones and Helm, 2009*; *Johnson, 2010*; *Oliynyk et al., 2010*; *Crowther et al., 2013*). As such, structural information on this class of synthetases will also facilitate bioengineering of improved analogs. We describe here the 2.2 Å resolution structure of CylM and demonstrate that its dehydration domain surprisingly has structural similarity with eukaryotic lipid kinases despite the absence of notable sequence homology. These findings may also have implications for the three eukaryotic homologs of lanthipeptide cyclases, one of which was recently shown to interact with mammalian target of rapamycin (mTOR) complex 2 (*Zeng et al., 2014*).

## Results and discussion

### Overall structure and in vitro activity of CylM

CylM and its substrate peptides $CylL_L$ and $CylL_S$ were expressed in *Escherichia coli* as hexahistidine-tagged proteins and purified by metal affinity chromatography. Incubation of CylM with $CylL_L$ or $CylL_S$ in the presence of $MgCl_2$ and adenosine triphosphate (ATP), and subsequent removal of the leader peptides by purified CylA, a serine protease of the cytolysin biosynthetic pathway (*Booth et al., 1996*), resulted in the desired number of dehydrations as determined by matrix-assisted laser-desorption time-of-flight mass spectrometry (MALDI-TOF MS) (*Figure 1—figure supplement 1*).

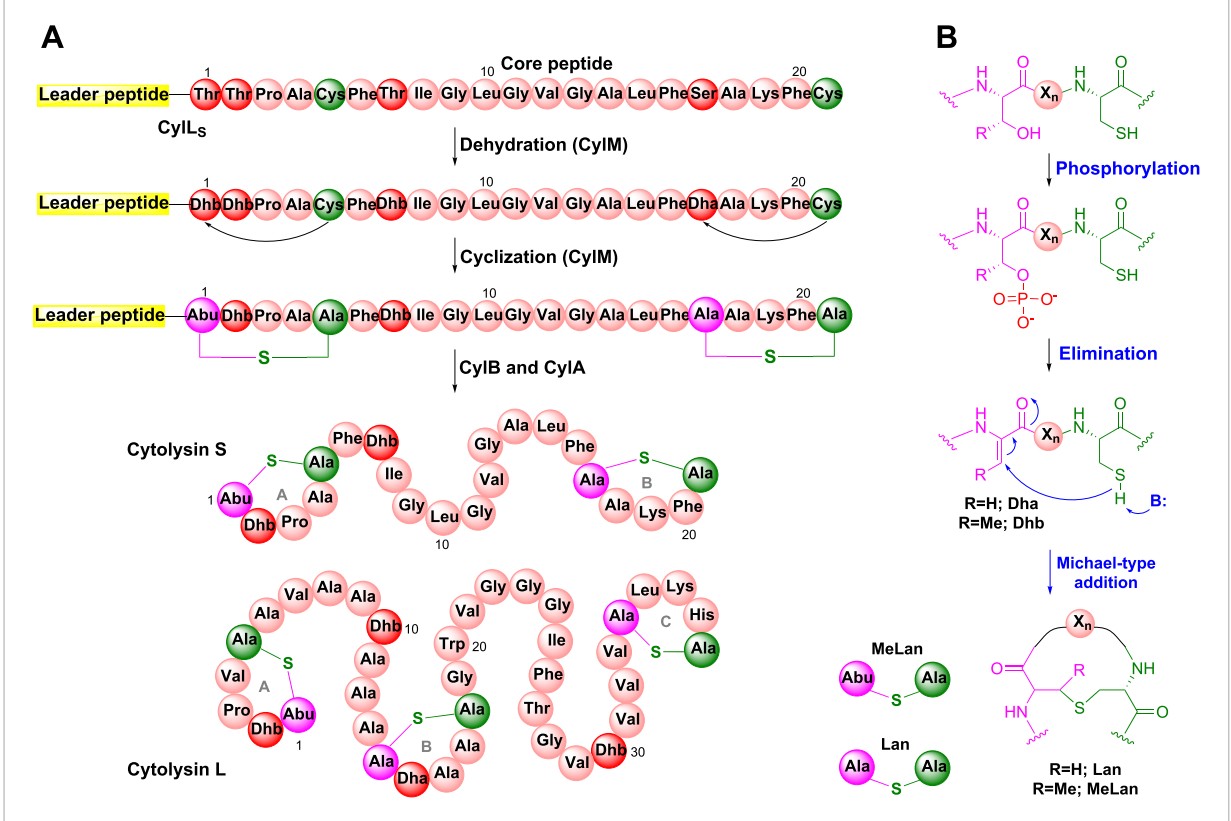

**Figure 1**. Biosynthesis of the enterococcal cytolysin. (**A**) Biosynthetic route to cytolysin S (small subunit of cytolysin) and the structure of cytolysin L (large subunit of cytolysin). CylM dehydrates three Thr and one Ser in the precursor peptide CylL$_S$ to generate three Dhb residues and one Dha. The enzyme also catalyzes the conjugate addition of the thiols of Cys5 to Dhb1 and Cys21 to Dha17. The proteases CylB and CylA then remove the leader peptide in a step-wise manner to provide cytolysin S. In similar fashion, CylM catalyzes seven dehydrations of Ser and Thr residues and three cyclization reactions during the biosynthesis of the large subunit of cytolysin. Abu-S-Ala = methyllanthionine (MeLan); Ala-S-Ala = lanthionine (Lan); Dha = dehydroalanine; Dhb = dehydrobutyrine. (**B**) Post-translational modifications carried out by CylM during cytolysin biosynthesis. X$_n$ = peptide linker.

The following figure supplements are available for figure 1:

**Figure supplement 1**. MALDI/TOF mass spectra for CylL$_L$ (**A**) and CylL$_S$ (**B**) peptides incubated with (magenta traces) or without (blue traces) CylM.

**Figure supplement 2**. ESI MS/MS analysis of CylL$_L$ (**A**) and CylL$_S$ (**B**) core peptides modified by CylM in vitro and treated with the protease CylA that removes the leader peptide.

Analysis of the peptides by tandem electrospray ionization mass spectrometry (ESI MS) demonstrated the formation of the correct ring structures (*Figure 1—figure supplement 2*).

To investigate the mechanism of catalysis, we determined the 2.2 Å resolution structure of CylM in complex with adenosine monophosphate (AMP). The structure of the ~110 kDa polypeptide consists of two distinct domains, with an N-terminal dehydration domain, composed of residues Asn4 through Pro624, and a C-terminal cyclization domain encompassed by Tyr641 through Glu992 (*Figure 2A*). The protein is a monomer in the crystal and in solution as determined by gel filtration analysis. Consistent with prior predictions, the cyclization domain consists of the α/α-barrel fold observed in the structure of the stand-alone class I Lan cyclase NisC (*Li et al., 2006*). As in NisC, CylM contains a single zinc ion near the center of the toroid coordinated by residues Cys875, Cys911, and His912, with a water molecule completing the tetrahedral coordination geometry at the metal. This zinc site is believed to activate the thiols of the Cys residues during the cyclization reaction (*Li et al., 2006*). The barrel of the NisC structure is interspersed with a structural element that resembles eukaryotic peptide-binding domains (*Figure 2B*), which is thought to bind the leader region of the substrate

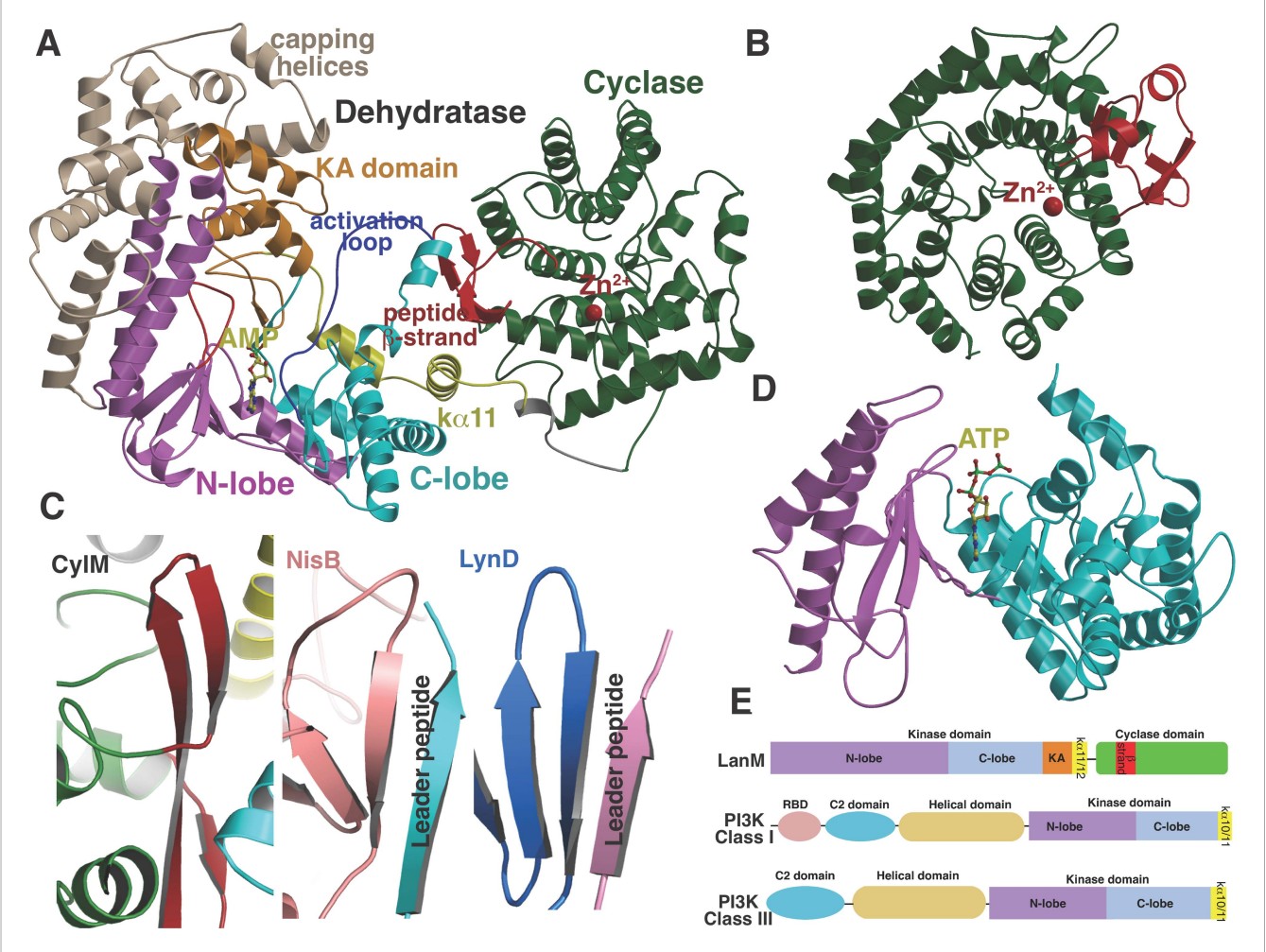

**Figure 2**. (**A**) Overall structure of CylM. (**B**) Structure of the class I lanthipeptide cyclase NisC illustrating the structural homology with the C-terminus of CylM. (**C**) Comparison of the putative peptide-binding β-strands of CylM with the peptide binding regions of other RiPP biosynthetic enzymes including NisB (involved in nisin biosynthesis, PDB 4WD9) and LynD (involved in cyanobactin biosynthesis; PDB 4V1T). (**D**) Structure of the lipid kinase PI3K that shares homology with the dehydration domain of CylM. (**E**) Domain organization of LanMs in comparison with that of lipid kinases. RBD, Ras-binding domain.

The following figure supplements are available for figure 2:

**Figure supplement 1**. MALDI-TOF mass spectrum for CylL$_S$ modified by the CylM dehydratase domain in *Escherichia coli*.

**Figure supplement 2**. Topology diagrams for (**A**) CylM and (**B**) PI3 kinase P110γ.

**Figure supplement 3**. Structure based alignment of biochemically characterized LanM enzymes.

peptide. The C-terminal cyclization domain of CylM lacks this element but instead contains a β-sheet region composed of three antiparallel strands that is situated near the zinc ion (*Figure 2A*, red). This element, encompassing Ile666 through Leu690, is located on the opposite face of the toroid relative to the putative leader peptide-binding domain in NisC, where it flanks against the base of the N-terminal dehydration domain of CylM. A similar antiparallel β-stranded element engages the leader peptide in the mechanistically unrelated class I Lan dehydratase NisB (*Ortega et al., 2015*) and is also found in other enzymes involved in the biosynthesis of ribosomally synthesized and post-translationally modified peptides (RiPPs) (*Koehnke et al., 2013*; *Burkhart et al., 2015*; *Koehnke et al., 2015*) (*Figure 2C*). Thus, the leader peptide binding architecture may be conserved across RiPP

biosynthetic enzymes, despite very high diversity of the reactions they catalyze (*Arnison et al., 2013*). To investigate whether the β-stranded element in the cyclase domain is important for the dehydration reaction, the N-terminal domain (residues 1–625) that lacks this element was expressed and purified with an N-terminal His$_6$-tag. Incubation with CylL$_S$ substrate resulted in efficient dehydration (*Figure 2—figure supplement 1*), indicating that the β-stranded element is not required for the dehydration reaction. This observation is consistent with several very recent reports describing expression and activity of the two individual domains of various LanM enzymes and binding of their substrates to both domains (*Ma et al., 2015*; *Shimafuji et al., 2015*; *Yu et al., 2015*).

Although the N-terminal dehydration domain lacks detectable sequence similarities with other proteins, the CylM structure reveals that this domain is architecturally related to the catalytic core of lipid kinases, such as phosphoinositide 3-kinase (PI3K) (*Figure 2A,D*) (*Walker et al., 1999*; *Williams et al., 2009*). The structural similarity with kinases is consistent with the proposed mechanism of dehydration by LanM proteins via first phosphorylation of Ser and Thr residues, followed by elimination of the phosphate (*Chatterjee et al., 2005*; *You and van der Donk, 2007*). Despite the structural homology, the topology of the CylM dehydration domain is quite different from those of canonical kinases (*Figure 2E*), resulting in a distinct connectivity between conserved secondary structural features (*Figure 2—figure supplement 2*), which can only be gleaned through a structure-based alignment (*Figure 2—figure supplement 3*).

## Structural details of the kinase and elimination active sites

Like canonical kinases, the CylM dehydration domain is composed of an N-lobe spanning residues Lys135 through Ser279, and a C-lobe composed of residues Glu280 through Val508 (*Figure 3A*). A number of helices formed by residues Leu5 through Asn131 and a two-helix insert created by Ile180 through Tyr200 cap the N-lobe, and hence we name these the 'capping helices' (*Figure 3A*). A similar, but topologically distinct, four-helix bundle insertion (termed the FRB domain [*Chen et al., 1995*]) cradles the N-lobe of PI3K-related protein kinases (PIKKs) such as, mTOR and DNA-PKc (*Figure 3B,C*) (*Choi et al., 1996*; *Sibanda et al., 2010*; *Yang et al., 2013*). Another distinct domain that we term the kinase-activation (KA) domain (see below) is held in place through interactions with the 'capping helices' (*Figure 3A*). The CylM C-lobe is appended with two helices formed by Gln589 through Pro624 that are characteristic of lipid kinases (helices kα10 and kα11 in lipid kinase nomenclature) (*Walker et al., 1999*; *Miller et al., 2010*; *Sibanda et al., 2010*; *Yang et al., 2013*). Helix kα11 of CylM packs against the three-β-stranded proposed leader peptide-binding region in the cyclase domain (*Figure 3A*). In mTOR, the equivalent kα11 helix is necessary for stabilization of the activation loop (*Figure 3B*) (*Yang et al., 2013*). As a result of all of these architectural additions, the CylM dehydration domain is considerably larger than the catalytic domain of other protein and lipid kinases, with the exception of the aforementioned PIKKs that contain the FRB insertion within the N-lobe (*Figure 3—figure supplement 1*) (*Sibanda et al., 2010*; *Yang et al., 2013*).

All protein and lipid kinases contain a requisite ∼30-residue segment termed the activation loop that plays a major role in both regulation and function. Kinase activity is typically controlled through activation-induced conformational changes, consisting of a disorder-to-order transition of the activation loop, which aligns active site residues and provides part of the binding site for substrate. Unlike most kinases, the activation loop in the CylM dehydration domain, composed of residues Asn365 through Val383, is well defined in the absence of bound peptide substrate (*Figures 2A, 3A*). The orientation and stabilization of the activation loop is established through numerous interactions with the LanM-specific KA domain, which is itself held in place through interactions with the 'capping helices' (*Figures 2A, 3A*). The activation loop is presumably stabilized in a catalytically competent conformation, because unlike most lipid kinases, the dehydration activity of LanM enzymes is not dependent on exogenous regulatory protein activators. The activation loop in mTOR is similarly held in a catalytically competent conformation via interactions with a highly conserved and integral ∼35 residue FATC domain, which is stabilized through packing interactions with a ∼40 residue insertion in the C-lobe termed the LBE (*Figure 3B*) (*Yang et al., 2013*).

## Substrate binding sites in CylM

In the CylM co–crystal structure, the bound AMP is located between the two lobes of the kinase domain (*Figure 3D*). The phosphate-binding loop (P-loop) is composed of residues Ser247 through

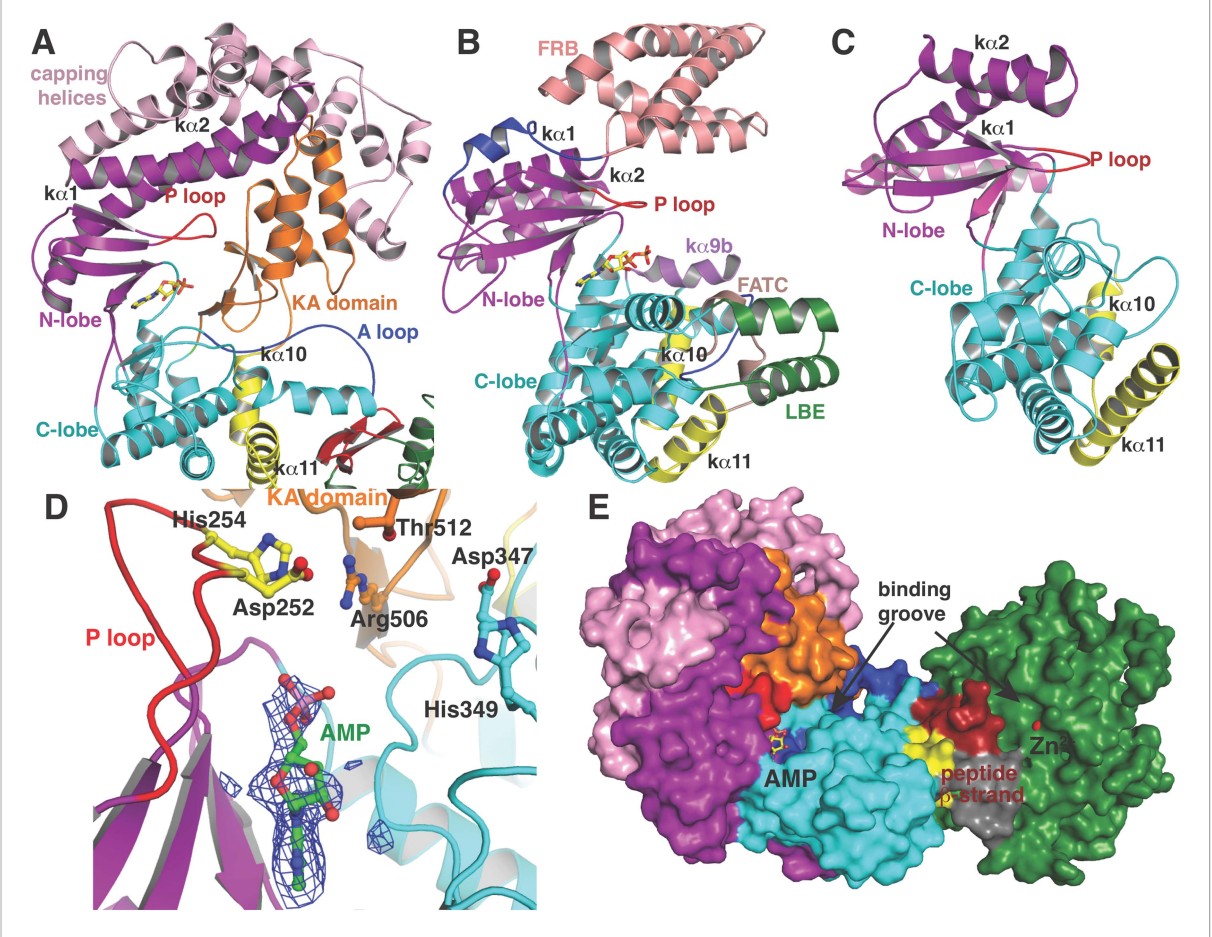

**Figure 3**. (**A–C**) Comparison of the kinase domains of (**A**) CylM, with those of (**B**) mammalian target of rapamycin (mTOR) and (**C**) DNA-PKc (a PI3 kinase). Secondary structural elements are colored as in *Figure 2A* and structurally unique insertions are designated. (**D**) Close up of the CylM dehydratase active site showing the bound nucleotide, and the proximity of residues important for phosphorylation and phosphate elimination. A simulated annealing difference Fourier map (calculated without the nucleotide) is superimposed in blue mesh. (**E**) Solvent occluded surface showing the two possible peptide-binding grooves that flank the peptide β-strand element (red). A loop = activation loop.

The following figure supplements are available for figure 3:

**Figure supplement 1**. Two views, rotated by 180°, of the superposition of the kinase active site of CylM (in purple) with the kinase domain of PI3 kinase (in cyan).

**Figure supplement 2**. Superposition of the active sites of CylM (pink) with the co-crystal structures of transition state mimics bound to mTOR (cyan) and cyclin-dependent protein kinase CDK2 (green).

**Figure supplement 3**. Superposition of the active sites of CylM (pink) with cyclin-dependent protein kinase CDK2 bound to a peptide substrate (green).

Thr262 and is considerably longer than the equivalent feature in PI3Ks. It contains many residues that are poised to interact with the nucleotide phosphate, including Asp252 and His254. Mutational analysis suggests a second role for these residues in the elimination of phosphate from the phosphorylated peptide product (see below). The adenine is situated in a hydrophobic binding pocket common across other structurally characterized kinases that is defined by CylM residues Val272 (Ile831 in PI3Kγ), Val301 (Tyr867 in PI3Kγ), Ile354 (Met953 in PI3Kγ), and Val361 (Phe961 in PI3Kγ). The proposed mechanism for PIKKs involves a conserved DxH motif, and LanM-conserved residues Asp347 and His349 are poised for catalysis in CylM (*Figure 3D*). His349 may receive a proton from Ser/Thr in the substrate peptide (*Miller et al., 2010*), in which case Asp347 likely orients the Ser/

Thr oxygen for nucleophilic attack onto the γ-phosphate of ATP. Alternatively, Asp347 could accept the proton from substrate (*Yang et al., 2013*). The invariant Lys residue that activates the γ-phosphate in kinases (*Hanks and Hunter, 1995*) is Lys274 in CylM. Additionally, two conserved residues, Asn352 and Asp364, are situated to act as divalent metal ligands to stabilize the incipient charge in the transition state, although slight reorientation of the side chain conformations must occur upon binding of the metal. A superposition of CylM with recent structures of the cyclin-dependent PIKK CDK2 (*Bao et al., 2011*) and mTOR (*Yang et al., 2013*), each bound to transition state mimics, reveals a near-perfect coincidence of equivalent residues at the active site, underscoring their importance in catalysis (*Figure 3—figure supplement 2*).

The structure also suggests a model for how the substrate peptide can bind to the two active sites. A superposition of CylM with the CDK2-substrate peptide bound structure (*Bao et al., 2011*) reveals that the LanM-specific KA domain occludes the canonical peptide binding sites of protein and lipid kinases (*Figure 3—figure supplement 3*). Instead, a solvent-excluded surface diagram demarcates a groove that leads to the nucleotide-binding site of the dehydratase domain (*Figure 3E*). A second groove traces to the zinc ion in the cyclase domain.

## Rate of ATP consumption in the presence of substrate peptide

In order to establish the functional relevance of the observed structural similarities between CylM and lipid kinases, we measured the kinetics for ATP hydrolysis by CylM in the presence of the substrate peptide CylL$_S$. Using a commercially available coupled luminescence assay kit that detects adenosine diphosphate (ADP), the steady-state kinetic parameters for ATP consumption by CylM were measured affording an apparent $K_M$ value of $99 \pm 6$ μM for ATP and a $k_{cat, app}$ of $4.1 \pm 0.1$ min$^{-1}$ (*Figure 4*); because poor solubility precluded saturation in the peptide substrate, these are apparent values. By way of comparison, prior studies established the kinetic parameters for the kinase domain of mTOR against the 4EBP1 peptide substrate yielding a $K_M$ of 9.5 μM for ATP and $k_{cat}$ of 0.91 min$^{-1}$ (*Tao et al., 2010*). Thus, the catalytic efficiency of ATP consumption by CylM is roughly within the same order of magnitude of the basal activity of the kinase domain of mTOR (which is enhanced ~fivefold in mTOR complex 1 [*Tao et al., 2010*]).

## Mutagenesis of potential catalytic residues involved in phosphorylation and phosphate elimination

Given the unanticipated lipid kinase fold, we focused our mechanistic studies on the CylM dehydration reaction. The aforementioned residues in the active site of CylM (Asp347, His349, Asn352, Asp364, and Lys274) are conserved in the LanM family. Their importance was investigated by replacement with Ala and assessing the activity with CylL, active site of$_S$ as substrate. No dehydration could not be detected for any of the mutants with the exception of CylM-H349A and K274A that both produced a small amount of dehydrated CylL$_S$ as determined by MALDI-TOF MS (*Figure 5* and *Figure 5—figure supplements 1, 2* and *Figure 5—source data 1–3*).

CylM is distinct from canonical PIKKs in that it not only phosphorylates its substrate but also eliminates the phosphate to generate a dehydroamino acid. Previous mutagenesis studies identified four conserved residues in LanMs that are important for the phosphate elimination reaction (*You and van der Donk, 2007*; *Ma et al., 2014*). Inspection of the CylM structure shows that these four residues (Asp252, His254, Arg506, and Thr512) are in close proximity to each other despite a separation of >250 amino acids in primary sequence, and that they are situated in or immediately adjacent to the phosphorylation site (*Figure 3D*). Their importance was investigated

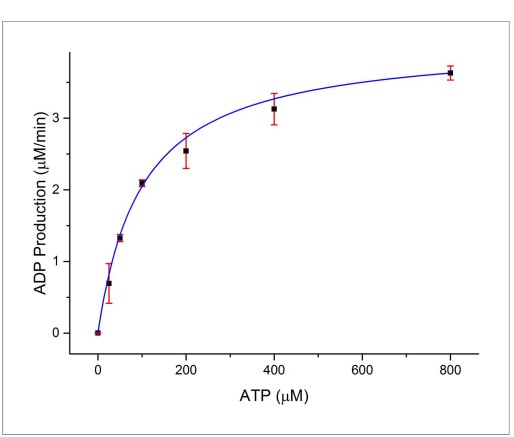

**Figure 4**. Dependence of the rate of ADP production by CylM (1 μM) on ATP concentration in the presence of 100 μM CylL$_S$.

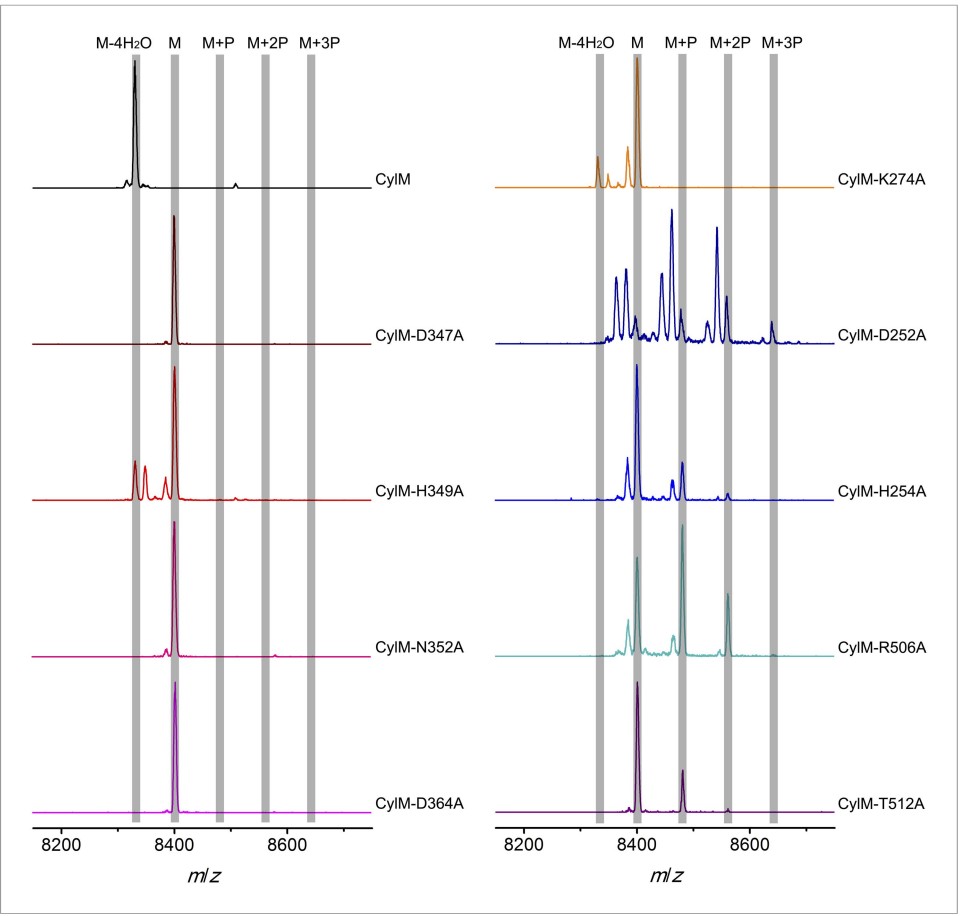

**Figure 5**. MALDI-TOF mass spectra of CylL$_S$ peptides co-expressed with CylM and CylM mutants in *E. coli*. M = unmodified CylL$_s$; P = phosphorylation. Peaks between the highlighted masses of multiply phosphorylated CylL$_S$ correspond to intermediates resulting from both phosphorylation and partial dehydration. A table showing the calculated and observed masses of each intermediate is provided in *Figure 5—source data 1*.

The following source data and figure supplements are available for figure 5:

**Source data 1**. Calculated and observed masses of CylL$_S$ peptides modified by CylM and CylM mutants *in E. coli*.

**Source data 2**. Calculated and observed masses of CylL$_S$ peptides incubated with CylM and CylM mutants in vitro for 30 min.

**Source data 3**. Calculated and observed masses of CylL$_S$ peptides incubated with CylM and CylM mutants in vitro for 10 hr.

**Figure supplement 1**. MALDI-TOF mass spectra of CylL$_S$ peptides incubated with CylM and CylM phosphorylation-deficient mutants in vitro for 30 min (left) and 10 hr (right).

**Figure supplement 2**. MALDI-TOF mass spectra of CylL$_S$ peptides incubated with CylM elimination-deficient mutants in vitro for 30 min (left) and 10 hr (right).

using alanine substitution. Phosphorylated intermediates were detected for all four mutants, with partially dehydrated products observed for all except CylM-T512A (*Figure 5* and *Figure 5—figure supplements 1, 2* and *Figure 5—source data 1–3*). Thus, all four residues are important for phosphate elimination. Asp252 and His254 are in the P-loop, which is thought to activate NTPs for attack during hydrolysis or substrate phosphorylation by interacting with the γ-phosphate. The mutant phenotypes suggest that these residues may play a similar role of phosphate stabilization during the phosphate

elimination reaction. Arg506 and Thr512 are located within the KA domain, suggesting that, in addition to stabilizing the activation loop, this domain also provides residues to assist in the elimination of phosphate. CylM thus offers insights into how an existing fold for an enzymatic activity (phosphorylation) can be adopted to carry out a second activity (elimination).

The mechanism to achieve dehydration in CylM is decidedly different from that found in other lanthipeptide synthetases. In class I, the Ser and Thr side chain hydroxyl groups are activated by glutamylation in a glutamyl-tRNA-dependent process (*Garg et al., 2013*; *Ortega et al., 2015*). Class III and IV lanthipeptide synthetases are made up of separate Ser/Thr protein kinase, phosphoSer/phosphoThr elimination, and cyclase domains that are readily recognized by sequence homology (*Goto et al., 2010*). The distinct phosphorylation and elimination domains in these latter enzymes require the phosphorylated peptides to translocate from the kinase to the lyase active site, accounting for the observation of phosphorylated intermediates (*Jungmann et al., 2014*). The adjacency of the phosphorylation and elimination active sites in CylM provides an explanation for the lack of observed phosphorylated substrate peptides as intermediates in LanM catalysis if elimination occurs faster than peptide dissociation (*Thibodeaux et al., 2014*).

To further investigate the elimination step, a mixture of phosphorylated CylL$_S$ peptides carrying different numbers of phosphate esters was obtained by co-expression of CylL$_S$ with the elimination-deficient CylM-R506A mutant in *E. coli*. The purified phosphorylated peptides were then incubated with wild-type CylM. Without addition of nucleotides, CylM did not eliminate the phosphates. However, when ADP or ADP analogs were supplied, the phosphorylated peptides were converted to dehydrated CylL$_S$ peptides (*Figure 6* and *Figure 6—figure supplements 1, 2*). Collectively, our results are consistent with an ordered kinetic mechanism in which ADP needs to bind before the phosphorylated peptide or in which the presence of ADP within the active site increases the affinity for phosphorylated peptide intermediates. The results are also consistent with processive phosphory-lation and elimination steps since ADP present in the active site from the phosphorylation reaction is required for the phosphate elimination reaction.

## Potential implications for eukaryotic lanthionine cyclase (LanC)-like proteins

The unexpected structural homology of bacterial LanM proteins with eukaryotic lipid kinases may also have implications for the function of three mammalian LanC-like (LanCL) proteins (*Chung et al., 2007*; *Sturla et al., 2009*; *Zhong et al., 2012*; *Huang et al., 2014*). LanC proteins are stand-alone Lan cyclases such as NisC (*Figure 2B*). Both LanCL1 and LanCL2 bind glutathione (*Chung et al., 2007*), with the thiol of glutathione coordinating the conserved zinc site in LanCL1 (*Zhang et al., 2009*). Hence, the human proteins also appear to activate a thiol, like the LanC proteins and the homologous C-terminal domains of the bacterial LanM enzymes. Although the precise functions of LanCL proteins are currently still unresolved, human LanCL1 has been shown to be important for antioxidant activity that is key to neuronal survival (*Huang et al., 2014*). Furthermore, recent studies indicated regulation of and physical interactions between the human LanCL2 and the kinases Akt and mTORC2 (*Zeng et al., 2014*). The structure of CylM shows that its LanC domain interacts with the activation loop and the kα11 helix of the kinase domain. These observations provide a platform to further investigate the intermolecular interaction of LanCL proteins with mammalian kinases, such as mTOR, that have structural homology with the CylM kinase domain.

## Conclusion

The structural and biochemical analysis of the lanthipeptide synthetase CylM provided here presents the first molecular picture for installation of the thioether crosslinks in the large family of class II lanthipeptides. Leader-dependent binding of the substrate would template movement of the core peptide between the dehydration and cyclization domains. The immediacy of the phosphorylation and phosphate-elimination sites allows for both reactions to occur in a processive manner to yield the dehydroamino residue, which can then be consigned to the cyclization domain for subsequent Michael-type addition reaction. The ordered activation loop in CylM precludes the need for an activation-induced conformational change, observed for other lipid kinases, as binding of the substrate is dictated largely by the leader sequence (*Abts et al., 2013*; *Thibodeaux et al., 2015*). The

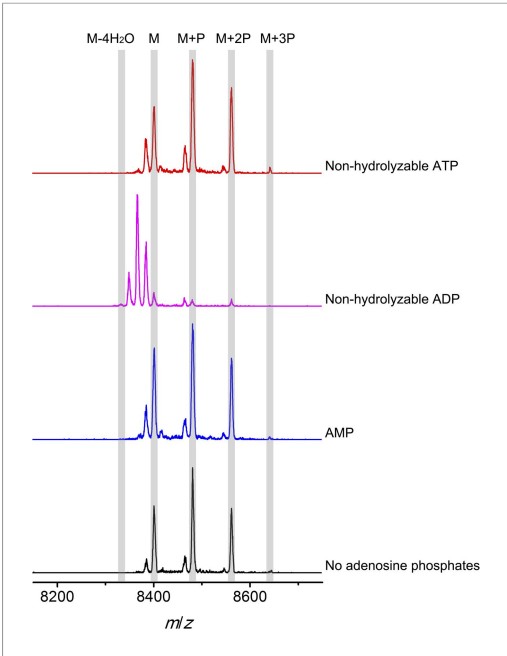

**Figure 6**. MALDI-TOF mass spectra of phosphorylated CylL$_S$ intermediates incubated with CylM in the absence of nucleotides (black trace), and in the presence of AMP (adenosine 5'-monophosphate disodium salt) (blue trace), non-hydrolyzable ADP (adenosine 5'-(β-thio)diphosphate trilithium salt) (magenta trace), or non-hydrolyzable ATP (adenosine 5'-(β,γ-imido)triphosphate lithium salt hydrate) (red trace). M = unmodified CylLs; P = phosphorylation. The data are shown for non-hydrolyzable analogs of ADP and ATP to distinguish whether the observed activity is due to the presence of these nucleotides, or to the activated phosphor-anhydride groups of ADP/ATP (see 'Materials and methods' for more information). See also *Figure 6—figure supplement 1*.

The following figure supplements are available for figure 6:

**Figure supplement 1**. MALDI-TOF mass spectra of CylL$_S$ peptides incubated with CylM in the absence of nucleotides (black trace), and in the presence of AMP (adenosine 5'-monophosphate disodium salt) (blue trace), non-hydrolyzable ADP (adenosine 5'-(β-thio)diphosphate trilithium salt) (magenta trace), or non-hydrolyzable ATP (adenosine 5'-(β,γ-imido)triphosphate lithium salt hydrate) (red trace).

**Figure supplement 2**. MALDI-TOF mass spectra of phosphorylated CylL$_S$ intermediates incubated with CylM in the absence (black trace) or presence of ADP (magenta trace).

structure of the CylM protein now allows installation of probes to monitor the movement of the substrates between the two active sites in LanM proteins to better understand the substantial motions of the substrate peptides during catalysis (*Thibodeaux et al., 2014*). In addition, it facilitates inhibitor design to prevent biosynthesis of the cytolysin virulence factor in pathogenic *E. faecalis*, one of the causative agents of vancomycin-resistant enterococcal infections (*Van Tyne et al., 2013*).

## Materials and methods

### General methods

The genes encoding CylM, CylL$_L$, and CylL$_S$ were synthesized by GeneArt (Invitrogen, Carlsbad, CA) with codon usage optimized for *E. coli* expression. All polymerase chain reactions were carried out on a C1000 thermal cycler (Bio-Rad, Hercules, CA). DNA sequencing was performed by ACGT, Inc (Wheeling, IL). Preparative HPLC was performed using a Waters Delta 600 instrument equipped with appropriate columns. LC-ESI-Q/TOF MS analyses were conducted using a Synapt G2 MS system equipped with Acquity UPLC (Waters, Milford, MA). MALDI-TOF MS was carried out on a Bruker Daltonics Ultra-fleXtreme MALDI-TOF/TOF mass spectrometer (Bruker, Billerica, MA). C18 zip-tip pipet tips were obtained from Millipore to desalt samples for MS analysis. Luminescence in 96-well plates was measured with a Synergy H4 Microplate Reader (BioTek, Winooski, VT).

Oligonucleotides were purchased from Integrated DNA Technologies (Coralville, IA). Restriction endonucleases, DNA polymerases, T4 DNA ligase, and media components were obtained from New England Biolabs (Ipswich, MA) and Difco laboratories (Franklin Lakes, NJ), respectively. Chemicals were ordered from Sigma Aldrich (St Louis, MI) or Fisher Scientific (Hampton, NH). The ADP-Glo MAX Assay kit was obtained from Promega (Madison, WI). *E. coli* DH5α and *E. coli* BL21 (DE3) cells were used as host for cloning and plasmid propagation, and host for expression, respectively. Expression vectors (pET15b and pRSFDuet-1) were obtained from Novagen (Billerica, MA).

### Cloning of *cylM*, *cylL$_L$*, and *cylL$_S$* genes into expression vectors

The *cylM* gene was cloned into the multiple cloning site 1 of a pRSFDuet-1 vector using *EcoR*I and *Not*I restriction sites to generate pRSFDuet-1/CylM plasmid. Primer sequences

Table 1. Primer sequences used for cloning of *cylM* and its mutants

| Primer name | Primer sequence (5'-3') |
|---|---|
| CylM_EcoRI_Duet_FP | AAAAA GAATTCG GAAGATA ATCTGATTAA T |
| CylM_NotI_Duet_RP | AAAAA GCGGCCGC TTACAGT TCAAACAGCA G |
| CylM_D252A_QC_FP | AGGGT GCA AGCCAT AGCCGTGGTAAAACCGTT AGC |
| CylM_D252A_QC_RP | ATGGCT TGC ACCCT GGC TTTCGCTAAT GCTATTCAGT |
| CylM_H254A_QC_FP | GATAGC GCT AGCCGT GGT AAAACCGTT AGCACCCTG |
| CylM_H254A_QC_RP | ACGGCT AGC GCTA TC ACCCTGGC TTTCGCTAAT G |
| CylM_D347A_QC_FP | GTTACC GCT CTGCAT TATGAAAACATCATTGCCCATGGC |
| CylM_D347A_QC_RP | AT GCAG AGC GGT AAC ATTAAAC AGAAAGGCAA TGCCAATCAG |
| CylM_H349A_QC_FP | CCGATCTG GCT TATGAAAA CATCATTGCCCATGGCGAATA |
| CylM_H349A_QC_RP | TTTTCATA AGC CAGATCGG T AACATTAAAC AGAAAGGCAA TGCCAAT |
| CylM_N352A_QC_FP | CATTATGAA GCC ATCATTGC CCATGGCGAATATCCG GTGATT |
| CylM_N352A_QC_RP | GCAATGAT GGC TTCATAATG CAGATCGGT AACATTAAAC AGAAAGGC |
| CylM_D364A_QC_FP | GTGATTATT GCT AATGAAACC TTTTTTCAGCAGAATATTCCGATTGAATTT |
| CylM_D364A_QC_RP | GGTTTC ATT AGC AATA ATCAC CGGAT ATTCGCCATG GGC |
| CylM_R506A_QC_FP | TGATTGTG GCC AATGTTAT TCGTCCGACCCAGCGTTA |
| CylM_R506A_QC_RP | A TAACATT GGC CACAATCA GA TTCTGCAGAT TATTATTAAT ATAGGCCAGA |
| CylM_T512A_QC_FP | GTCCG GCC CAG C GTTATGCAGATATGCTGGAA TTTAGC |
| CylM_T512A_QC_RP | CTG GGCCGGAC GAA TAACATTGCG CACAATCAGA |
| CylM_NdeI_FP | AAAAA CATATG GAAGATA ATCTGATTAA T |
| CylM625_KpnI_RP | AAAAA GGTACC TTA GTACGGGTTA TAAATATTCA G |

used are listed in *Table 1*. CylL$_L$ and *cylL$_S$* genes were cloned into a pET15b vector using *NdeI* and *BamHI* restriction sites, resulting in pET15b/CylL$_L$ or pET15b/CylL$_S$ plasmids, respectively.

## Construction of pRSFDuet-1 derivatives for expression of CylM mutants and for co-expression of CylL$_S$ with CylM mutants

The plasmids pRSFDuet-1/CylM-D347A, pRSFDuet-1/CylM-H349A, pRSFDuet-1/CylM-N352A, pRSFDuet-1/CylM-D364A, pRSFDuet-1/CylM-D252A, pRSFDuet-1/CylM-H254A, pRSFDuet-1/CylM-R506A, pRSFDuet-1/CylM-T512A, pRSFDuet-1/CylL$_S$/CylM-D347A-2, pRSFDuet-1/CylL$_S$/CylM-H349A-2, pRSFDuet-1/CylL$_S$/CylM-N352A-2, pRSFDuet-1/CylL$_S$/CylM-D364A-2, pRSFDuet-1/CylL$_S$/CylM-D252A-2, pRSFDuet-1/CylL$_S$/CylM-H254A-2, pRSFDuet-1/CylL$_S$/CylM-R506A-2 and pRSFDuet-1/CylL$_S$/CylM-T512A-2 were generated using QuikChange methodology using pRSFDuet-1/CylM and pRSFDuet-1/CylL$_S$/CylM-2 as templates, respectively (*Tang and van der Donk, 2013*). Primer sequences are listed in *Table 1*.

## Construction of the pRSFDuet-1/CylL$_S$/CylM-1-625-2 plasmid

The *cylM-1-625* gene was amplified and cloned into the MCS2 of pRSFDuet-1/CylL$_S$ to generate pRSFDuet-1/CylL$_S$/CylM-1-625-2. Primer sequences are listed in *Table 1*.

## Expression and purification of His$_6$-CylL$_L$ and His$_6$-CylL$_S$ peptides

*E. coli* BL21 (DE3) cells were transformed with pET15b/CylL$_L$ or pET15b/CylL$_S$ and plated on a LB plate containing 100 mg/l ampicillin. A single colony was picked and grown in 20 ml of LB in the presence of ampicillin at 37℃ for 12 hr. The cell suspension was directly used to inoculate 2 l of fresh LB media. Cells were cultured at 37℃ until the OD at 600 nm reached 0.5, and isopropyl β-D-1-thiogalactopyranoside (IPTG) was added to a final concentration of 0.2 mM. Cells were cultured at 37℃ for another 3–5 hr before harvesting. The cell pellet was resuspended at room temperature in LanA start buffer (20 mM NaH$_2$PO$_4$, 500 mM NaCl, 0.5 mM imidazole, 20% glycerol, pH 7.5 at 25℃)

and lysed by sonication. The resulting sample was then centrifuged at 23,700×$g$ for 30 min and supernatant was discarded. The remaining pellet was resuspended in LanA buffer 1 (6 M guanidine hydrochloride, 20 mM $NaH_2PO_4$, 500 mM NaCl, 0.5 mM imidazole, pH 7.5 at 25°C) and sonicated. Centrifugation was performed afterwards to pellet the debris and the soluble portion was passed through 0.45-μm syringe filters. His-tagged peptides were purified by immobilized metal ion affinity chromatography (IMAC) eluting with LanA elute buffer (4 M guanidine hydrochloride, 20 mM $NaH_2PO_4$, 500 mM NaCl, 1 M imidazole , pH 7.5 at 25°C). The eluted fractions were desalted by preparative HPLC using a Waters Delta-pak C4 column (15 μm 300 Å 25 × 100 mm). The resulting peptides were lyophilized to dryness and kept at −20°C for future use.

## Expression and purification of His$_6$-CylM and CylM mutants

*E. coli* BL21 (DE3) cells were transformed with pRSFDuet-1/CylM, pRSFDuet-1/CylM-D347A, pRSFDuet-1/CylM-H349A, pRSFDuet-1/CylM-N352A, pRSFDuet-1/CylM-D364A, pRSFDuet-1/CylM-D252A, pRSFDuet-1/CylM-H254A, pRSFDuet-1/CylM-R506A or pRSFDuet-1/CylM-T512A, and plated on a LB plate containing 50 mg/l kanamycin. A single colony was picked and grown in 20 ml of LB in the presence of kanamycin at 37°C for 12 hr. The cell suspension was directly used to inoculate 2 l of LB and cells were cultured at 37°C until the OD at 600 nm reached 0.5. The culture was cooled down on ice followed by the addition of IPTG to a final concentration of 0.1 mM. Cells were cultured at 18°C for additional 18 hr before harvesting. The harvested cells were resuspended on ice in LanM start buffer (20 mM HEPES, 1 M NaCl, pH 7.5 at 25°C) and lysed using a homogenizer. Insoluble debris was removed by centrifugation at 23,700×$g$ for 45 min at 4°C and the supernatant was passed through 0.45-μm syringe filters. His-tagged proteins were purified by IMAC, eluting with a linear concentration gradient of imidazole from 30 mM to 200 mM. The eluted fractions were analyzed using SDS-PAGE. Fractions containing the desired protein were combined and concentrated using a centrifugal filtering device, and the buffer was exchanged to LanM start buffer using a gel-filtration column. Protein concentration was quantified by its absorbance at 280 nm. The extinction coefficient for His$_6$-CylM was calculated as 140,110 $M^{-1} cm^{-1}$. Aliquoted protein solutions were flash-frozen and kept at −80°C for further usage.

## Expression and purification of His$_6$-CylL$_S$ peptides co-expressed with CylM mutants

*E. coli* BL21 (DE3) cells were transformed with pRSFDuet-1/CylL$_S$/CylM-D347A-2, pRSFDuet-1/CylL$_S$/CylM-H349A-2, pRSFDuet-1/CylL$_S$/CylM-N352A-2, pRSFDuet-1/CylL$_S$/CylM-D364A-2, pRSFDuet-1/CylL$_S$/CylM-D252A-2, pRSFDuet-1/CylL$_S$/CylM-H254A-2, pRSFDuet-1/CylL$_S$/CylM-R506A-2, pRSFDuet-1/CylL$_S$/CylM-T512A-2, or pRSFDuet-1/CylL$_S$/CylM-1-625-2, and plated on a LB plate containing 50 mg/l kanamycin. A single colony was picked and grown in 10 ml of LB in the presence of kanamycin at 37°C for 12 hr. The cell suspension was directly used to inoculate 1 l of LB and cells were cultured at 37°C until the OD at 600 nm reached 0.5. The culture was cooled down on ice followed by the addition of IPTG to a final concentration of 0.1 mM. Cells were cultured at 18°C for 18 hr before harvesting. To obtain both fully modified and linear CylL$_S$ as well as possible intermediates (partially modified CylL$_S$) and reduce the bias introduced by peptide solubility, harvested cells were resuspended and lysed directly in LanA buffer 1 (6 M guanidine hydrochloride, 20 mM $NaH_2PO_4$, 500 mM NaCl, 0.5 mM imidazole, pH 7.5 at 25°C) by sonication. Debris was removed by centrifugation and the soluble portion was passed through 0.45-μm syringe filters. His-tagged CylL$_S$ was purified by IMAC, eluting with LanA elute buffer (4 M guanidine hydrochloride, 20 mM $NaH_2PO_4$, 500 mM NaCl, 1 M imidazole, pH 7.5 at 25°C). The eluted fractions were desalted with Strata-X polymeric reverse phase SPE columns and lyophilized to dryness.

## Reconstitution of CylM activity in vitro

Presumably due to high hydrophobicity, the solubility of linear CylL$_L$ or CylL$_S$ peptides is extremely poor. For enzyme assays, 2 mg/ml peptide suspension was made in deionized water as stock solution for both peptides. The stock solution was vortexed to a homogenized suspension each time before any peptide was taken. However, given the presence of precipitation, the concentration of CylL$_L$ or CylL$_S$ peptides could not be tightly controlled.

To reconstitute the activity of CylM in vitro, 20 µM of linear peptides were supplied in a reaction vessel with 4 mM MgCl$_2$, 2 mM ATP, 2 mM DTT, $1 \times 10^{-5}$ U thrombin (to remove the His-tag in situ) and 50 mM HEPES (pH 7.5), followed by the addition of CylM to a final concentration of 0.5 µM. Reactions were incubated at room temperature for 4 hr. Control reactions were set up with all other components in the absence of CylM. Each sample was zip-tipped and analyzed by MALDI-TOF MS. Aliquoted samples were treated by CylA (serine protease encoded in the biosynthetic pathway of cytolysin) to remove the leader peptides, and the resulting core peptides were analyzed by LC-MS or LC-MS/MS.

ESI MS analysis confirmed a mass shift of 144 or 126 Da for CylL$_L$, corresponding to a loss of 8 or 7 water molecules, with the 7-dehydrated peptide as the major product, and a mass shift of 72 Da for CylL$_S$, corresponding to a loss of 4 water molecules (*Figure 1—figure supplement 1*). The results were consistent with the reported mass of CylL$_L$' and CylL$_S$' as well as our previous observations using an *E. coli* co-expression system, where 7 dehydrations and 4 dehydrations were detected (*Booth et al., 1996*; *Tang and van der Donk, 2013*). Tandem MS (MS/MS) analysis indicated the desired ring systems were formed for both peptides (*Figure 1—figure supplement 2*).

## In vitro modification of CylL$_S$ by CylM mutants

CylL$_S$ peptide (20 µM) was supplied to a reaction vessel in the presence of 4 mM MgCl$_2$, 2 mM ATP, 2 mM DTT, $1 \times 10^{-5}$ U thrombin (to remove the His-tag in situ) and 50 mM HEPES (pH 7.5). CylM and CylM mutant proteins were then added to a final concentration of 0.5 µM. Reactions were incubated at room temperature and aliquots were quenched by adding formic acid to a final concentration of 0.5% at desired time points. Each sample was then zip-tipped and analyzed by MALDI-TOF MS.

With linear CylL$_S$ serving as the substrate, wild-type CylM finished the modification by eliminating 4 water molecules within 30 min of incubation when characterized using MALDI-TOF MS. In comparison, the four phosphorylation-deficient mutants were unable to convert the starting material into modified peptide using the same set up (*Figure 5—figure supplement 1*). A small amount of dehydrated product was observed for CylM-H349A, which afforded partially dehydrated intermediates when analyzed using the *E. coli* co-expression system (*Figure 5—figure supplement 1*). We further increased the incubation time to 10 hr at room temperature to facilitate the detection of any minimal level activity. Indeed, almost full modification of CylL$_S$ was achieved by CylM-H349A (*Figure 5—figure supplement 1*). Partially modified products with one dehydration and one phosphorylation were also detected for CylM-N352A, but CylL$_S$ remained unmodified in the presence of CylM-D347A and CylM-D364A even with elongated incubation time (*Figure 5—figure supplement 1*).

In vitro characterization of the four elimination-deficient mutants of CylM also provided similar phenotypes as what was observed using the co-expression system, except that CylM-H254A afforded fully modified CylL$_S$ after elongated incubation period (*Figure 5—figure supplement 2*), indicating that mutating histidine 254 to alanine slows down but does not abolish the phosphate-elimination activity of CylM. The T512A mutant did not eliminate the phosphate even with increased reaction time (*Figure 5—figure supplement 2*), suggesting that Thr512 is critical for the elimination activity of CylM.

## Elimination activity of CylM in the presence of adenosine derivatives

Phosphorylated CylL$_S$ peptides carrying different numbers of phosphate esters were obtained by co-expression of His$_6$-CylL$_S$ with CylM elimination-deficient mutant CylM-R506A. The IMAC-purified peptide mixture was dissolved in deionized water to make a 350 µM stock solution. Non-hydrolyzable ATP (adenosine 5′-(β,γ-imido)triphosphate lithium salt hydrate), non-hydrolyzable ADP (adenosine 5′-(β-thio)diphosphate trilithium salt) and AMP (adenosine 5′-monophosphate disodium salt) were reconstituted in deionized water and a stock solution of 20 mM was obtained for each. For elimination reactions, CylM was present at a final concentration of 0.5 µM in the presence of 1 mM MgCl$_2$, 2 mM DTT and 50 mM HEPES (pH 7.5). Then adenosine derivatives (final concentration 500 µM) or deionized water (negative control) were added, followed by phosphorylated CylL$_S$ peptide to a final concentration of 35 µM, and the assay was incubated at room temperature for 2 hr. Parallel control reactions were set up using linear CylL$_S$ with a peptide concentration of 20 µM in the presence of $1 \times 10^{-5}$ U thrombin (to remove the His-tag in situ). Samples were zip-tipped and analyzed by MALDI-TOF MS (*Figure 6—figure supplement 1*).

Non-hydrolyzable adenosine derivatives were used for analysis of the elimination activity because we determined that CylM could use both ATP and ADP to dehydrate its substrates (i.e., both ATP and

ADP can be used for phosphorylation). Hence, when the mixture of CylL$_S$ peptides that carry 1–3 phosphate esters were supplied to CylM in the presence of ATP or ADP, both elimination and dehydration reactions proceeded, which complicated the outcome and precluded data interpretation. For example, when ADP was supplied instead of non-hydrolyzable ADP, only fully (fourfold) dehydrated CylL$_S$ was observed (*Figure 6—figure supplement 2*). Since the phosphorylated peptides carried only 1–3 phosphate esters, the additional dehydrations resulted from conversion of non-phosphorylated Ser/Thr to Dha/Dhb. Therefore, to study the elimination reaction in isolation, non-hydrolyzable ATP and ADP analogs were used.

## Overexpression, purification, and crystallization of CylM

Single colonies of chemically competent *E. coli* Rosetta 2 cells, transformed with the pRSFDuet-1/CylM plasmid, were grown in LB media supplemented with kanamycin (50 µg/ml) and chloramphenicol (25 µg/ml). A 6 ml starter culture was grown overnight and used to inoculate 1 l of LB media supplemented with the same antibiotic. Liquid cultures were grown at 37°C with vigorous shaking, and protein production was induced with the addition of 0.5 mM IPTG when the OD600 reached 0.5 followed by further shaking for additional 20 hr at 18°C and 200 rpm. Cell pellets were harvested from the cultures by centrifugation at 4°C, followed by suspension of the pellet in ~30 ml of buffer (500 mM NaCl, 10% glycerol, 20 mM Tris, pH 8.0). Frozen cell pellets were thawed and lysed by sonication, and the lysates were clarified by centrifugation at 4°C. The clear supernatant containing the soluble fraction was loaded onto a 5 ml immobilized metal ion affinity resin column (Hi-Trap Ni-NTA, GE Healthcare) pre-equilibrated with binding buffer (1 M NaCl, 5% glycerol, 20 mM Tris, pH 8.0). The column was washed with 50 ml of 12% elution buffer (1 M NaCl, 250 mM imidazole, 20 mM Tris, pH 8.0), and eluted by a linear gradient. Fractions containing the highest purity protein, as judged by Coomassie-stained SDS-PAGE, were pooled and further purified by size exclusion chromatography (Superdex Hiload 200 16/60, GE Healthcare) in 500 mM KCl, 20 mM HEPES, pH 7.5 buffer. The purified protein was concentrated using Amicon Ultra-4 centrifugal filters (10 KDa molecular weight cut-off, Millipore) and stored in liquid nitrogen until needed. The final concentration was quantified by Bradford analysis (Thermo Scientific).

Crystals of LanM were obtained by hanging drop vapor diffusion method, by mixing 1 µl of protein (concentration of 2–6 mg/ml) with an equal volume of precipitant of either 0.2 M CaCl, 0.1 M HEPES pH 7.5, 10 mM betaine hydrochloride, and 28% PEG 400 (condition 1) or 0.2 M KCl, 0.05 M HEPES pH7.5, 10 mM barium chloride, and 33% 5/4 PO/OH (condition 2). Crystals were supplemented with either PEG 400 or 5/4 PO/OH to a final concentration of 35% (vol/vol) prior to vitrification by direct immersion in liquid nitrogen. Macro- and micro-seeding facilitated the formation of crystals suitable for diffraction data collection. SeMet CylM was expressed, purified, and crystallized in a similar manner. Native and SeMet data were collected at Sector 21 ID (LS-CAT, Advanced Photon Source, Argonne National Labs, IL) and data were integrated and scaled using HKL2000 (*Otwinowski et al., 2003*) or XDS (*Kabsch, 2014*). Crystallographic phases were determined by single wavelength anomalous diffraction methods from data collected on crystals of SeMet CylM to a resolution limit of 2.7 Å. Heavy atom sites were located using the SHELX (*Sheldrick, 2010*) suite of programs and refinement of heavy atom parameters in SHARP (*Bricogne et al., 2003*) yielded an initial figure of merit of 0.273. Multiple rounds of automated and manual model building using COOT (*Emsley et al., 2010*), interspersed with rounds of crystallographic refinement using REFMAC5 (*Murshudov et al., 2011*), resulted in convergence to the near final model (free R factor of 0.30). Ligand and water molecules were added at this stage and refinement was completed using BUSTER (*Blanc et al., 2004*). The validity of all models was routinely determined using MOLPROBITY (*Chen et al., 2010*) and by using the free R factor to monitor improvements during building and crystallographic refinement. Relevant data collection, phasing, and refinement statistics may be found in *Table 2*.

## ADP production by CylM

Stock solutions of 800 µM ATP and ADP were prepared by diluting the Ultra Pure ATP and ADP supplied with the ADP-Glo MAX Assay kit (Promega) in 1× reaction buffer (20 mM MgCl$_2$, 2 mM DTT, and 50 mM HEPES pH 7.5). Mixtures of 50 µl of ATP (800 µM) and ADP were made in which the final ADP content varied from 0 to 20% (0, 8, 16, 24, 32, 40, 80, and 160 µM), which represents the percent conversion of ATP to ADP in the kinetic experiments. These standard solutions were designated the

**Table 2**. Data collection, phasing, and refinement statistics

|  | Native | SeMet |
|---|---|---|
| Data collection |  |  |
| Space group | $P2_12_12_1$ | $P2_12_12_1$ |
| Unit cell: a, b, c (Å) | 51.2, 90.7, 246.4 | 51.2, 90.9, 246.2 |
| Resolution (Å)* | 50.00–2.2 (2.24–2.2) | 50.00–2.8 (2.85–2.8) |
| Total reflections | 359,303 | 169,660 |
| Unique reflections | 58,180 | 25,354 |
| $R_{sym}$ (%) | 6.3 (67.0) | 6.1 (53.8) |
| $I/\sigma(I)$ | 19.1 (1.7) | 16.7 (2.1) |
| Completeness (%) | 97.9 (87.6) | 96.0 (88.4) |
| Redundancy | 6.2 (5.1) | 6.0 (5.9) |
| Refinement |  |  |
| Resolution (Å) | 25.0–2.2 |  |
| No. reflections used | 51,874 |  |
| $R_{work}/R_{free}$‡ | 23.7/26.8 |  |
| Number of atoms |  |  |
| Protein | 7251 |  |
| Solvent | 160 |  |
| Metal/Nucleotide | 1/23 |  |
| B-factors |  |  |
| Protein | 52.9 |  |
| Solvent | 32.4 |  |
| Metal/Nucleotide | 54.1/62.3 |  |
| R.m.s deviations |  |  |
| Bond lengths (Å) | 0.011 |  |
| Bond angles (°) | 1.54 |  |

*Highest resolution shell is shown in parenthesis.
‡R-factor = $\Sigma(|F_{obs}| - k|F_{calc}|)/\Sigma |F_{obs}|$ and R-free is the R value for a test set of reflections consisting of a random 5% of the diffraction data not used in refinement.

800 µM series. Mixtures in a 400 µM series were prepared by diluting 25 µl of the 800 µM series samples with 25 µl 1× reaction buffer. Similarly, 200, 100, 50, and 25 µM series (25 µl each) were prepared. To each sample, 25 µl of the ADP-Glo Reagent was added and incubated for 40 min, followed by the addition of 50 µl of ADP-Glo Max Detection Reagent and incubation for 1 hr. All samples were then transferred into a white 96 well plate and the luminescence was measured by a plate reader. Standard curves (25, 50, 100, 200, 400, and 800 µM series) were created to correlate the ADP concentration with the luminescence.

To measure the ATP consumption of CylM in vitro, 25 µl of reaction mixtures were prepared consisting of 100 µM linear peptide, 1 µM CylM, and 1× reaction buffer, followed by the addition of ATP to final concentrations of 25, 50, 100, 200, 400, and 800 µM. Control reactions were set up with all other components in the absence of CylM. Reactions were incubated at 25°C for 1, 2, 3, and 4 min before being stopped by adding 25 µl of ADP-Glo Reagent, which depletes the remaining ATP. After incubation for 40 min, ADP-Glo Max Detection Reagent (50 µl) was then added, the samples were incubated for 1 hr, and the luminescence was measured. ADP production in each sample was calculated by applying the corresponding standard curve. The curve of ATP consumption of CylM against ATP concentration was fitted using OriginPro 2015. All reactions were carried out in duplicate.

## Acknowledgements

This work was supported by grants from the National Institutes of Health (R01 GM058822 to WAvdD and RO1 GM079038 to SKN). A Bruker UltrafleXtreme MALDI TOF/TOF mass spectrometer was purchased in part with a grant from the National Institutes of Health (S10 RR027109 A).

## Additional information

### Competing interests

WAD: Reviewing editor, *eLife.* The other authors declare that no competing interests exist.

### Funding

| Funder | Grant reference | Author |
|---|---|---|
| National Institutes of Health (NIH) | R01 GM058822 | Wilfred A van der Donk |
| National Institutes of Health (NIH) | R01 GM079038 | Satish K Nair |

The funder had no role in study design, data collection and interpretation, or the decision to submit the work for publication.

### Author contributions

S-HD, Performed all crystallographic experiments., Conception and design, Acquisition of data, Analysis and interpretation of data; WT, Designed, analyzed, and performed all biochemical assays and helped write the manuscript, Conception and design, Acquisition of data, Analysis and interpretation of data, Drafting or revising the article; TL, TL helped collect the X-ray data, Acquisition of data; YY, Developed the method to generate just the kinase domain of CylM, Conception and design; SKN, Designed and analyzed all crystallographic data and helped write the manuscript, Conception and design, Analysis and interpretation of data, Drafting or revising the article; WAD, Designed and analyzed the biochemical experiments and helped write the manuscript., Conception and design, Analysis and interpretation of data, Drafting or revising the article

### Author ORCIDs

Wilfred A van der Donk, http://orcid.org/0000-0002-5467-7071

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
