## [Decision Letter]

Thank you for submitting your work entitled “The enterococcal cytolysin synthetase has an unanticipated lipid kinase fold” for peer review at *eLife*. Your submission has been favorably evaluated by Richard Losick (Senior Editor), a Reviewing Editor, and two reviewers.

The reviewers have discussed their reviews with one another, and the Reviewing Editor has drafted this decision to help you prepare a revised submission.

Two reviewers have read your manuscript and they both found its findings to be interesting and significant. The reviewers have also provided some useful comments for you to consider during revision. Specifically, they struggled to visualize the structural relatedness between CylM and lipid kinases as presented in Figure 3. It therefore seems important for you to improve this section of the manuscript and its presentation. In complementary ways, both reviewers also raised concerns about whether CylM truly exhibits kinase-like activities that would support the claim that structural homology implies functional homology. The reviewers have suggested some specific experiments to address this point, as well as to test whether the role of ADP in the CylM mechanism might be more than just structural. Hopefully, you can perform these additional experiments to more directly evaluate the kinase-like functions of CylM and the potential role of the ADP product in the second phosphate elimination step. Some of the specific comments of the reviewers are included below.

Summary of reviews:

The article by Dong et al. reveals the first structural information on a LanM lanthipeptide synthetase. The authors reveal that the enzyme contains a dehydratase domain that shares homology with lipid kinases. The authors provide a strong rationale for how the enzyme carries out a concerted phosphorylation, and elimination reaction. The van der Donk laboratory has a strong background in the study of these enzymes, and this structural work provides a strong underpinning to much of both theirs and others’ work in the field. The work is overall of high quality. However, there are a few points that need to be addressed before the work is ready for publication.

Important revisions:

1) The authors do not show in the work the rates of ATP and peptide substrate turnover by the enzyme, and it would be interesting to see how effective this enzyme is compared to other protein and lipid kinases. The authors note that the activation loop appears to be maintained in an ordered active conformation. It would be useful to measure turnover of ATP to confirm that CylM indeed has reaction rates comparable to other protein or lipid kinases. A simple experiment could be devised to measure the rates of ATP to ADP turnover, and compare this to other lipid or protein kinases.

2) The authors show detailed structural comparisons between CylM and PI3Kγ in Figure 3. The authors describe this in words; however, in the figure it is very difficult to make out the insertions specific to CylM (specifically the overlay presented in Figure 3). It would potentially be useful to see PI3K, CylM, and mTOR side by side to see the insertions present in CylM compared to PI3K, and its similarity to mTOR. Additionally, due to the use of multiple colors and the large format representation of secondary structure, it is very difficult to visualize the structural similarities in Figure 3. The representation would benefit by superimposing the two structures using narrow loops (not ribbons) of small width, and in only two colors with proper depth queuing, in order to visualize the backbone.

3) One fascinating element of this manuscript was the activity dependence of ADP by the WT phosphate elimination. The authors include the use of non-hydrolyzable ADPγS, but offer little more discussion. Why didn't the authors use ADP alone? What other nucleotide analogs are sufficient for this reaction? Is the role merely structural (as implied) or could there be a catalytic element? Given that ADP is the product of the kinase 1/2 reaction, this special relationship deserves more thorough investigation.

Minor points:

1) The discussion of LanC at the end of the Discussion seems to be premature. An alignment of the mTOR kinase domain with LanC would be essential to underscore if the LanC domain would be compatible for binding. As a side note the authors do not define LanC as the cyclase domain, and clarification in the wording of the text would be useful for readers outside of the field.

2) The authors have carried out very detailed analysis of substrates and phosphate ester analogs for both the wild type and mutant enzymes. What are the side products shown in Figure 4 for the mutants CylM D252A, and L247A? It would be interesting to see if these are potential trapped intermediates, or if these are contaminants from a given protein preparation.

3) For many lipid kinases it has been difficult to crystallise enzymes in the presence of nucleotide analogs. It would be useful to generate an omit map for AMPPNP, to give readers confidence in the occupancy of this pocket.

---

## [Author Response]

*1) The authors do not show in the work the rates of ATP and peptide substrate turnover by the enzyme, and it would be interesting to see how effective this enzyme is compared to other protein and lipid kinases. The authors note that the activation loop appears to be maintained in an ordered active conformation. It would be useful to measure turnover of ATP to confirm that CylM indeed has reaction rates comparable to other protein or lipid kinases. A simple experiment could be devised to measure the rates of ATP to ADP turnover, and compare this to other lipid or protein kinases*.

A kinetic analysis of ATP consumption by CylM in the presence of its substrate CylL_S_ was performed according to the reviewer’s suggestion. We confirmed that CylM exhibits kinase-like activities with an apparent *K*_M_ value of 99 µM for ATP and an apparent k_cat_ of 4.08 min^‒1^. The solubility of the substrate does not allow determination of its kinetic constants, but the kinetic parameters varying the ATP concentration are of the same magnitude of what were reported for the kinase domain of mTOR in the presence of the 4EBP1 peptide substrate. We felt that that was the best kinase for comparison since it is a lipid kinase-like enzyme but its substrates are proteins/peptides (like CylM).

We believe these new data and the additional paragraph we added into the manuscript provides the reader with an idea regarding the overall kinetics of the kinase activity of CylM. Of course, the enzyme catalyzes more than just the phosphorylation (also elimination and cyclization) and it does so at multiple sites on its substrate. Hence, a detailed comparison with dedicated protein kinases is not readily accomplished. Nevertheless, the ordered loop observed in CylM does not seem to accelerate ATP hydrolysis significantly with this particular set up. We appreciate the suggestion of performing such an experiment.

*2) The authors show detailed structural comparisons between CylM and PI3Kγ in*
Figure 3*. The authors describe this in words; however, in the figure it is very difficult to make out the insertions specific to CylM (specifically the overlay presented in*
Figure 3*). It would potentially be useful to see PI3K, CylM, and mTOR side by side to see the insertions present in CylM compared to PI3K, and its similarity to mTOR. Additionally, due to the use of multiple colors and the large format representation of secondary structure, it is very difficult to visualize the structural similarities in*
Figure 3*. The representation would benefit by superimposing the two structures using narrow loops (not ribbons) of small width, and in only two colors with proper depth queuing, in order to visualize the backbone*.

We edited the original figures to show PI3K, CylM, and mTOR side by side (now Figure 3) for a better comparison. Superimposed structures of CylM and PI3K with narrow loops are now provided in Figure 3—figure supplement 1 according to the reviewers’ suggestion. Superposition of the kinase active sites of CylM, mTOR, and CDK2 are now provided as new Figure 3—figure supplement 2. We appreciate the suggestions that have indeed improved the visualization of the text.

*3) One fascinating element of this manuscript was the activity dependence of ADP by the WT phosphate elimination. The authors include the use of non-hydrolyzable ADPγS, but offer little more discussion. Why didn't the authors use ADP alone? What other nucleotide analogs are sufficient for this reaction? Is the role merely structural (as implied) or could there be a catalytic element? Given that ADP is the product of the kinase 1/2 reaction, this special relationship deserves more thorough investigation*.

We added a paragraph in the Materials and methods section to explain why we used non-hydrolyzable analogs. In short, we discovered that CylM can use both ATP and ADP to phosphorylate its substrates, although ATP is preferred over ADP. However, the use of ADP for phosphorylation presented a problem when we were trying to isolate just the elimination reaction.

The substrates used for the latter activity was a mixture of CylL_S_ peptides that carry different numbers of phosphate esters. When ADP was used to investigate the ability to eliminate the phosphates (as we described, no elimination is observed in the absence of nucleotides) elimination was observed, but the number of net dehydrations exceeded the number of initial phosphorylations. This is because new phosphorylations and eliminations were catalyzed by CylM in the presence of ADP. This unexpected observation complicated the outcome and made data interpretation difficult. For example, when ADP was supplied instead of non-hydrolyzable ADP using a similar set up, only fully dehydrated CylL_S_ could be observed after incubation. We added new data in Figure 6—figure supplement 2 to illustrate this. Therefore, to study the elimination reaction in isolation, non-hydrolyzable ATP and ADP analogues were employed to minimize the effect resulted from CylM-catalyzed phosphorylation.

*Minor points*:

*1) The discussion of LanC at the end of the Discussion seems to be premature. An alignment of the mTOR kinase domain with LanC would be essential to underscore if the LanC domain would be compatible for binding. As a side note the authors do not define LanC as the cyclase domain, and clarification in the wording of the text would be useful for readers outside of the field*.

We now defined LanCL in the heading of the section and added a few sentences explaining how LanC/LanCL/LanM are related. We also removed the speculation about possible evolutionary origins and removed the statement that the CylM structure may be a preview of the interactions of mammalian LanCL proteins with kinases. We did leave in a discussion of the previously observed interaction between mTORC1 and LanCL1, since it was found well before the structure of CylM was determined. We believe that the CylM structure does provide a potential platform to use for investigation of the mTOR-LanCL2 interaction and that pointing this out is useful for the pertinent community who may not necessarily recognize the relevance of a LanM structure.

*2) The authors have carried out very detailed analysis of substrates and phosphate ester analogs for both the wild type and mutant enzymes. What are the side products shown in*
Figure 4
*for the mutants CylM D252A, and L247A? It would be interesting to see if these are potential trapped intermediates, or if these are contaminants from a given protein preparation*.

They are CylM-modified CylL_S_ peptides carrying both phosphate esters and partial dehydrations. The presence of such intermediates indicates that the elimination activity of the CylM mutant was reduced but not completely abolished. We added mass tables with calculated and observed masses for each intermediate for a clearer presentation and we added a sentence to the legends explaining what these peaks are.

*3) For many lipid kinases it has been difficult to crystallise enzymes in the presence of nucleotide analogs. It would be useful to generate an omit map for AMPPNP, to give readers confidence in the occupancy of this pocket*.

A simulated annealing difference Fourier map (calculated without the nucleotide) was generated and shown in Figure 3. We appreciate the suggestion of adding such a figure.